# Direct mosquito feedings on dengue-2 virus-infected people reveal dynamics of human infectiousness

**Louis Lambrechts**[1]☺*, **Robert C. Reiner, Jr**[2]☺*, **M. Veronica Briesemeister**[3], **Patricia Barrera**[3,4], **Kanya C. Long**[5], **William H. Elson**[3], **Alfonso Vizcarra**[3], **Helvio Astete**[6,7], **Isabel Bazan**[6], **Crystyan Siles**[6], **Stalin Vilcarromero**[6¤a], **Mariana Leguia**[4], **Anna B. Kawiecki**[8], **T. Alex Perkins**[9], **Alun L. Lloyd**[10], **Lance A. Waller**[11], **Uriel Kitron**[12], **Sarah A. Jenkins**[6¤b], **Robert D. Hontz**[6¤c], **Wesley R. Campbell**[6¤d], **Lauren B. Carrington**[13], **Cameron P. Simmons**[14], **J. Sonia Ampuero**[6], **Gisella Vasquez**[7], **John P. Elder**[15], **Valerie A. Paz-Soldan**[16], **Gonzalo M. Vazquez-Prokopec**[12], **Alan L. Rothman**[17], **Christopher M. Barker**[8], **Thomas W. Scott**[3‡], **Amy C. Morrison**[8‡]

1 Institut Pasteur, Université Paris Cité, CNRS UMR2000, Insect-Virus Interactions Unit, Paris, France, 2 University of Washington, Seattle, Washington, United States of America, 3 Department of Entomology and Nematology, University of California, Davis, California, United States of America, 4 Genomics Laboratory, Pontificia Universidad Católica del Peru, Lima, Peru, 5 Department of Family Medicine and Public Health, University of California San Diego School of Medicine, La Jolla, California, United States of America, 6 Virology and Emerging Infections Department, United States Naval Medical Research Unit No. 6, Lima, Peru, 7 Department of Entomology, United States Naval Medical Research Unit No. 6, Lima, Peru, 8 Department of Pathology, Microbiology, and Immunology, School of Veterinary Medicine, University of California, Davis, California, United States of America, 9 Department of Biological Sciences and Eck Institute for Global Health, University of Notre Dame, Notre Dame, Indiana, United States of America, 10 Biomathematics Graduate Program and Department of Mathematics, North Carolina State University, Raleigh, North Carolina, United States of America, 11 Department of Biostatistics and Bioinformatics, Rollins School of Public Health, Emory University, Atlanta, Georgia, United States of America, 12 Department of Environmental Sciences, Emory University, Atlanta, Georgia, United States of America, 13 Global Malaria Program, World Health Organization, Geneva, Switzerland, 14 Institute for Vector-Borne Disease, Monash University, Clayton, Victoria, Australia, 15 School of Public Health, San Diego State University, San Diego, California, United States of America, 16 Department of Tropical Medicine, Tulane University School of Public Health and Tropical Medicine, New Orleans, Louisiana, United States of America, 17 Institute for Immunology and Informatics and Department of Cell and Molecular Biology, University of Rhode Island, Providence, Rhode Island, United States of America

☺ These authors contributed equally to this work.
¤a Current address: Hospital Nacional Edgardo Rebagliati Martins, ESSALUD Lima, Peru
¤b Current address: Naval Medical Research Command Silver Spring, Maryland, United States of America
¤c Current address: Department of Emerging Infectious Diseases, United States Naval Medical Research Unit TWO, Singapore, Singapore
¤d Current address: Department of Medicine, Walter Reed National Military Medical Center, Bethesda, Maryland, United States of America
‡ TWS and ACM also contributed equally to this work.
* louis.lambrechts@pasteur.fr (LL); bcreiner@uw.edu (RCR)

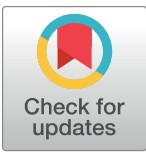

**Data Availability Statement:** R code and data used for statistical analysis and to produce figures in the manuscript are openly available on GitHub [https://

## Abstract

Dengue virus (DENV) transmission from humans to mosquitoes is a poorly documented, but critical component of DENV epidemiology. Magnitude of viremia is the primary determinant of successful human-to-mosquito DENV transmission. People with the same level of viremia, however, can vary in their infectiousness to mosquitoes as a function of other factors that remain to be elucidated. Here, we report on a field-based study in the city of Iquitos,

github.com/bcreiner/mosq_feeds_Iquitos] and provided as a supplementary file (S1 File), respectively.

**Funding:** The study was primarily funded by a grant from the U.S. National Institutes of Health/National Institute of Allergy and Infectious Diseases (P01 AI098670 to TWS) with some additional salary support provided by U01AI151814 to ACM. The study was also supported by the European Union's Horizon 2020 research and innovation programme under ZikaPLAN (734584 to LL) and the French Government's Investissement d'Avenir program Laboratoire d'Excellence Integrative Biology of Emerging Infectious Diseases (ANR-10-LABX-62-IBEID to LL). Further support was provided by a sub contract from the Bill and Melinda Gates Foundation to the University of Notre Dame (OPP1081737 to TWS/ACM), the Defense Threat Reduction Agency, Military Infectious Disease Research Program (MIDRP, S0520_15_LI and S0572_17_LI), and the Armed Forces Health Surveillance Division, Global Emerging Infections System Branch (ProMIS ID: 20160390169, P0090_17_N6_1.1.1, P0106_18_N6_01.01, and P0143_19_N6). The views expressed in this article are those of the authors and do not necessarily reflect the official policy or position of the Department of the Navy, Department of Defense, nor the U.S. Government. The funders had no role in study design, data collection and analysis, decision to publish, or preparation of the manuscript.

**Competing interests:** The authors have declared that no competing interests exist.

Peru, where we conducted direct mosquito feedings on people naturally infected with DENV and that experienced mild illness. We also enrolled people naturally infected with Zika virus (ZIKV) after the introduction of ZIKV in Iquitos during the study period. Of the 54 study participants involved in direct mosquito feedings, 43 were infected with DENV-2, two with DENV-3, and nine with ZIKV. Our analysis excluded participants whose viremia was detectable at enrollment but undetectable at the time of mosquito feeding, which was the case for all participants with DENV-3 and ZIKV infections. We analyzed the probability of onward transmission during 50 feeding events involving 27 participants infected with DENV-2 based on the presence of infectious virus in mosquito saliva 7–16 days post blood meal. Transmission probability was positively associated with the level of viremia and duration of extrinsic incubation in the mosquito. In addition, transmission probability was influenced by the day of illness in a non-monotonic fashion; i.e., transmission probability increased until 2 days after symptom onset and decreased thereafter. We conclude that mildly ill DENV-infected humans with similar levels of viremia during the first two days after symptom onset will be most infectious to mosquitoes on the second day of their illness. Quantifying variation within and between people in their contribution to DENV transmission is essential to better understand the biological determinants of human infectiousness, parametrize epidemiological models, and improve disease surveillance and prevention strategies.

## Author summary

In this study, we examined the potential for people with mild illness to transmit dengue virus to mosquitoes. Although people with mild dengue illness can be infectious, they often are undetected by surveillance systems and research teams. Variation in infectiousness over the course of a person's infection is largely unexplored, but potentially important component of dengue epidemiology. In an effort to better understand the factors that underlie this variation, we blood fed laboratory-reared, uninfected mosquitoes directly on mildly ill people naturally infected with dengue virus in Iquitos, Peru. Detection of virus in mosquito saliva after feeding on an infected person was considered positive for mosquito transmission. As has been reported previously, transmission to mosquitoes was positively associated with the amount of virus in a person's blood and the number of days after the mosquito fed on an infected person. Unlike results from previous studies, which reported peak transmission during the first day of a person's illness, we found that for mildly ill people infectiousness to mosquitoes peaked on the second day of illness. Because people with mild illness are thought to contribute more to dengue virus transmission than more severely ill people, subtle differences in the dynamics of their infectiousness to mosquitoes could be epidemiologically significant.

## Introduction

Dengue is one of the fastest-growing global infectious diseases [1]. It is now endemic in an increasing number of tropical cities and its geographical range is expected to further expand due to ongoing global phenomena including urbanization and climate change [2–4]. There are an estimated 400 million yearly human dengue virus (DENV) infections, of which approximately 100 million are apparent with symptoms [5]. Dengue is a self-limited infection with a

broad range of clinical manifestations ranging from asymptomatic infection to life-threatening illness [6]. Although people without any clinical manifestations or with mild symptoms can be infectious, they usually go undetected by the healthcare systems and thus constitute a considerable challenge to disease surveillance and control [7,8]. Understanding variation within and between people in their contribution to onwards virus transmission is essential to improve the theory of DENV epidemiology, lead to innovation in outbreak detection and response, and refine assessments of prevention strategies [9].

The mosquito, *Aedes aegypti*, is the principal DENV vector and a primary driver of the increasing disease burden and expanding geographic distribution of dengue [1,10]. After an *Ae. aegypti* female takes a blood meal from a viremic human, virus transmission can happen if the virus successfully infects and replicates in the mosquito's midgut epithelium, crosses the midgut basal lamina, infects and replicates in the salivary glands, and is released in saliva as the mosquito bites another person. This process is generally represented by two distinct epidemiological parameters. Vector competence refers to the physiological ability of a mosquito to become infected and subsequently transmit the virus, whereas the extrinsic incubation period (EIP) represents the typical duration of this process [11]. It is well established that DENV vector competence is positively correlated with the virus dose ingested in the blood meal [7,12–16]. Human infectiousness to mosquitoes roughly follows the kinetics of viremia, and symptomatic people can be infectious to mosquitoes from two days before to six days after the onset of symptoms [7,12–14,16,17]. Although human-to-mosquito DENV transmission is primarily determined by the magnitude of viremia, laboratory experiments indicate that vector competence and EIP also vary as a function of other factors such as the mosquito genotype, the virus strain, and the ambient temperature [18–23].

Several studies point to human factors other than viremia that influence DENV infectiousness to mosquitoes. A hospital-based study in Vietnam found that rising levels of DENV-specific antibodies were associated with a reduced likelihood of human-to-mosquito transmission independently of viremia [16]. A community-based study in Cambodia reported that asymptomatic and presymptomatic people were more infectious to mosquitoes than symptomatic people at the same level of viremia [7]. In laboratory studies, low-density lipoproteins [24,25] and serum iron [26] were shown to inhibit DENV acquisition by *Ae. aegypti* whereas blood glucose [27] and DENV non-structural protein 1 (NS1) antigenemia [28] enhanced mosquito infection. The blood concentration of some of these metabolites are known to vary within and between DENV-infected people [29,30] and could contribute to variation in DENV infectiousness to mosquitoes.

Regardless of the underlying mechanisms, knowledge of DENV transmission from humans to mosquitoes is epidemiologically imprecise and potentially misleading because it does not account for such variation in infectiousness within and between different people. This knowledge gap is due in large part to the logistical challenge of quantifying human-to-mosquito DENV transmission in a natural setting [31]. Here, we conducted a field-based study to characterize the temporal dynamics of viremic human infectiousness to mosquitoes with an emphasis on the difficult-to-study people with mild disease. Even though people with mild disease make up the bulk of human DENV infections, they are often missed by public health surveillance systems because surveillance programs are designed to identify infections in people who are ill enough to seek medical treatment [5]. We analyzed a dataset newly generated from a study design that we previously developed to identify DENV-infected volunteers with mild illness and then carried out direct mosquito feedings on those individuals [9,15,32]. Although our study focus was dengue, we also enrolled Zika virus (ZIKV)-infected participants during the study period when ZIKV circulated in Iquitos.

## Methods

### Ethics statement

The study protocol was approved by the U.S. Naval Medical Research Unit No. 6 (NAMRU-6) Institutional Review Board (Protocol Number NAMRU6.2014.0028), which includes Peruvian representation and complies with U.S. Federal and Peruvian regulations governing the protection of human subjects. Institutional Review Board authorization agreements were established between the U.S. NAMRU-6 and all participating institutions. The protocol was reviewed and approved by the Loreto Regional Health Department, which oversees health research in Iquitos. Written informed consent was obtained from all adult participants and the parents or guardians of all the child participants between 5–17 years. Participants that were minors provided assent (written for 8- to 17-years old, verbal for <8 years old).

### Study design and sample testing

**Study area and period.**   Iquitos is an urban center of ~400,000 inhabitants located in the Amazon Basin of northeastern Peru that functions almost as a self-contained epidemiological system because access to it is limited to air and river travel. The study area was previously described in detail [9]. DENV is endemic in Iquitos with a strong seasonal pattern. In general, DENV transmission is low from May to July, increases in August and September, and usually peaks between November and January. Our study was conducted from June 2015 to April 2019. In May 2016, the first human ZIKV infections were reported in Iquitos. ZIKV transmission continued through April 2017 with almost no detection of DENV during that time period.

**Study protocol.**   The overall study design and procedures are described in detail by Morrison et al. [9]. Briefly, DENV and ZIKV cases were identified from two ongoing community-based cohort studies (~20,000 people with 3–5 febrile illness surveillance visits per week) initiated in June 2015 and from passive clinic-based disease surveillance carried out daily in approximately 6 facilities [33]. People identified with active DENV and ZIKV infections (i.e., detectable viremia) were invited to participate. Participation consisted of (1) direct mosquito feeding, (2) collection of venous blood samples, (3) health monitoring, and (4) symptom and movement questionnaires [9]. Study procedures were repeated daily if the participant's blood from the previous day tested positive for DENV or ZIKV and the participant agreed to continue. Participants could decline individual procedures on individual days and remain in the study.

**Participant testing.**   Blood samples were generally tested within 24 hours of collection by reverse transcription quantitative PCR (RT-qPCR) for DENV and ZIKV RNA (see sample testing section below). As soon as results were available, study subjects who tested positive for DENV or ZIKV and had indicated willingness to participate in the viremic participant protocol were contacted and enrolled to perform the mosquito feeding and sequential blood sample collection procedures. During the ZIKV transmission period, Institutional Review Board permission was obtained to initiate the ZIKV-positive case protocol prior to obtaining the initial RT-qPCR result to maximize the probability of detecting human-to-mosquito transmission, because ZIKV-positive participants tended to be afebrile and were identified when they developed a rash. Participants were informed of their infection status within 12–24 hours and virus-positive participants were invited to repeat the mosquito feeding, blood sampling, and questionnaire procedures. Laboratory results were provided to study participants with an explanation by study personnel.

**Mosquitoes.**   Human infectiousness was experimentally assessed using laboratory-reared mosquitoes derived from the local, wild-type *Ae. aegypti* population. To minimize

uncontrolled differences between experiments and maintain a level of genetic diversity representative of the wild mosquito population, a genetically diverse laboratory strain (GDLS) was created as previously described [9,34,35]. Briefly, the GDLS was established between August and December 2015 by collecting ~1,000 immature *Ae. aegypti* at each of 10 locations across Iquitos. Wild female progenitors were screened by RT-qPCR for DENV, the only known *Aedes*-borne virus circulating in Iquitos at the time, before hatching their eggs. None of the field-caught progenitors tested positive for DENV RNA. The 10 strains were maintained separately and merged to generate the GDLS mosquitoes used in blood-feeding experiments. All experiments were conducted within the first 6 generations of the GDLS. Virus-free mosquitoes were maintained in a field insectary constructed in a local household with a relative humidity (RH) of 70–80% and a temperature of approximately 27–28°C. The insectary had windows that let in natural light and room lights were turned on from approximately 0800 to 1700 each day. The natural photoperiod in Iquitos averages 12 hour:12 hour light:dark and variation between minimum and maximum daylight is approximately 1 hour. Mosquitoes that fed on viremic participants were held in a secure, laboratory-based insectary under the same temperature, RH, and light conditions. The GDLS mosquitoes were maintained with weekly non-infectious blood meals (chicken blood available for sale from a local butcher) through an artificial membrane. Resulting eggs were hatched in a water/tea (Collins Cinnamon tea) infusion (1 liter) for 24 hours. Two-hundred larvae per liter of tap water were placed in white 26.5 × 16.5 cm plastic pans. Larvae were provided a combination of wheat powder mixed with commercial fish food daily until pupation. Pupae were transferred to 1-pint plastic containers and placed into 1-gallon cardboard cages, provided with a moist towel to maintain humidity, a sugar cube, and a water pledget on top of the cages. As the mosquitoes emerged, females were separated into 1-pint cardboard cages containing 25 female *Ae. aegypti* each and 5 males to allow for mating. The workflow was adjusted to ensure that sufficient mosquitoes to carry out serial feeding experiments on up to 5 participants per week (60 mosquitoes per feeding event × average of 3 feeding events per participant × 5 participants = 900 per week). Sugar was removed from half of the cages on alternate days to ensure availability of sugar-starved mosquitoes on any given day.

**Blood feeding.** Mosquitoes were blood fed directly on the viremic participants according to a previously established procedure [9,15,32]. Briefly, two containers of 30 (15 for children <14 years of age) *Ae. aegypti* females from the GDLS were placed on the arms or legs of the participant for a maximum of 10 minutes in their homes. The option to carry out the feeding experiment in the U.S. Naval Medical Research Unit No. 6 (NAMRU-6) laboratory was offered as an alternative. Containers were then transported in a secure container to the NAMRU-6 laboratory for processing. Containers were placed in a −20°C freezer for about 1 minute to immobilize mosquitoes. Unfed mosquitoes were removed, and blood-engorged mosquitoes were kept in 1-pint cardboard containers with mesh tops, provided a sugar cube and sucrose solution soaked pledget, and held at 27°C and 70–80% RH for 7 to 16 days in a secure NAMRU-6 insectary. Mosquitoes were collected early (7–8 days) or late (12–16 days) relative to an expected EIP of about 10 days at 27°C [36]. The duration of extrinsic incubation was assigned randomly to containers. For most feeding events, mosquitoes were divided so that some individuals were tested early (7–8 days) and others from the same feeding event were tested later (12–16 days).

**Vector competence.** On the day of harvest, saliva was collected from ~20 blood-fed mosquitoes according to a previously described protocol [16] (see Fig 3 in [9]). Briefly, legs and wings were removed, and the proboscis was inserted into the end of a pipette tip containing a 1:1 solution of fetal calf serum and 30% sucrose. After 30 minutes, the solution in the pipette tip was immediately inoculated intrathoracically into 4 to 5 naïve (i.e., not previously blood-

fed) recipient *Ae. aegypti*. After 7 days of incubation, inoculated mosquitoes that were still alive (typically 3–5) were harvested to extract total RNA. Recipient mosquitoes that shared a saliva inoculum from the same mosquito were pooled for RNA extraction. Aliquots of RNA extracted from each of the ~20 inoculated mosquito pools were combined and tested for DENV or ZIKV RNA by RT-qPCR. If the pool of RNA extracts was negative, the participant was scored not infectious to mosquitoes for that feeding event. If the pool of RNA extracts was positive, RNA from each of the ~20 inoculated mosquito pools was individually tested by RT-qPCR to determine the proportion of blood-fed mosquitoes with infectious saliva. If multiple feedings were performed on a single participant, each feeding event was considered independent of each other and pools were tested as described above.

**Sample testing.** Viral RNA was detected and quantified by RT-qPCR. RNA was extracted from whole blood and/or serum samples using QIAamp Viral RNA Mini Kit (Qiagen) following the manufacturer's guidelines. The same extraction method was used for all mosquito samples except for inoculated mosquitoes, from which DENV RNA was extracted using a crude extraction method previously described [37]. Briefly, mosquito samples were placed in a vial containing a 4-mm glass bead and homogenized with a TissueLyser (Qiagen) after adding 250 μl of squash buffer (10 mM Tris pH 8.2, 1 mM EDTA, 50 mM NaCl) and 15 mg/ml of proteinase K (Qiagen). The homogenate was then heated at 56°C for 5 minutes and 98°C for 10 minutes. DENV RNA was detected and quantified using a serotype-specific TaqMan RT-qPCR assay based on a previously published method [38]. ZIKV RNA was detected and quantified using aTaqMan RT-qPCR assay targeting the envelope gene (forward primer: 5'- GAT CCTACTGCTATGAGGCATC-3'; reverse primer: 5'- CCAGCCTCTGTCCACTAACGT-3'; probe: 5'- FAM-AGCCTACCTTGACAAGCAGTCAGACACTCAA-BHQ-3') according to the following thermal profile: 50°C for 15 minutes, 95°C for 2 minutes, followed by 40 cycles of 95°C for 15 seconds and 58°C for 30 seconds. RT-qPCR assays were performed on the ABI 7500 real-time PCR platform (Applied Biosystems) using TaqMan Fast Virus 1-step RT-PCR master mix (Life Technologies) or AgPath-ID One-Step RT-PCR kit (Applied Biosystems) for DENV and Superscript III Platinum One-Step qRT-PCR kit (Invitrogen) for ZIKV. Human samples with borderline cycle threshold (Ct) values for DENV (Ct = 34–36) were confirmed positive or negative using previously extracted RNA and a nested RT-PCR protocol previously described [39]. For ZIKV, samples with borderline Ct values underwent repeat testing. The limits of detection for the DENV and ZIKV assays were 12 and 31 genome copies/μl, respectively.

## Data analysis

**General model considerations.** Details of the model selection process are provided in S1 Text. All candidate regression models relating the probability of an individual blood-fed mosquito becoming infectious to covariates were generalized linear models or generalized linear mixed models, specifically binomial regressions with a logit link. All models were run in R 4.1.3 [40]. The potential functional forms of the relationships that were evaluated, depending on the covariate, were (a) linear, (b) binary, (c) non-parametric, (d) smooth non-linear, or (e) shape-constrained smooth non-linear. Working backwards from the most complex models, any regression that contained a shape-constrained smooth term (see Viremia section below for details) was evaluated using mosquito-level data (i.e., presence or absence of virus in saliva as determined by inoculation and subsequent testing of recipient mosquitoes) and fit using the shape-constrained additive model (scam) package [41]. In this, the response variable was thus 0 (virus-negative saliva) or 1 (virus-positive saliva). All other models were evaluated using summarized experiment-level data; i.e., the fraction of tested mosquitoes that had virus-

positive saliva for a particular participant-incubation combination. As such, the response variable was equivalent to $k$ infectious mosquitoes out of $n$, where $k$ and $n$ were specific to a particular participant-incubation combination. Any model that did not contain a shape-constrained term, but did contain a smooth non-linear term (e.g., the potential for a non-linear relationship between day of illness and transmission probability) was fit using generalized additive models with the `mgcv` package [42]. Models that had no smooth terms, but did consider repeated observations from the same participant, were fit using generalized linear mixed models with a random effect by individual with the `lme4` package [43].

**Treatment of covariates.** Investigations were initiated with the simplest model and the most complex model and increased or decreased complexity as a function of model fit, parsimony, and biological feasibility, with the goal to arrive at a single model that balanced these three components. For each covariate considered, the simplest and most complex models considered are detailed below, as well as, in one case, justification for constraining the functional form to adhere more closely to biological plausibility. When possible (i.e., when comparing nested models), model comparison was done using a likelihood-ratio test. Otherwise, when appropriate, models were compared using Akaike information criterion (AIC) [44]. It should be noted that due to different response variables, models fit with `scam` were not directly comparable to other models in terms of AIC. In many cases, however, the same model could be reformulated in `scam` to allow for direct comparison. In terms of parsimony, when two compared models had similar AIC values (within 2 units of each other [45]), the model without the shape constraint was selected over the model with one, and the linear model was selected over the non-linear one. Also in the interest of parsimony, all non-linear smooth terms were constructed with 3 interior knots and all shape-constrained smooth terms were constructed with 4 interior knots; the minimum allowed in the respective packages.

**Viremia.** Consistent with earlier similar studies [7,16], viremia was measured as viral RNA concentration in serum samples and $\log_{10}$ transformed. The simplest relationship considered between viremia and the probability of a mosquito becoming infectious was linear. *A priori* there is an expectation that a higher viremia would result in a higher probability of transmission; this is an assumption supported by results reported in previous literature [7,16]. As such, for the most complex relationship considered, the potential for a shape-constrained relationship between viremia and the response variable was considered, specifically a non-decreasing, monotonic smooth relationship.

**Day of illness.** At the time of mosquito feeding, the number of days after symptom onset ranged from 1 to 5 with more than 10 observations for each of the first 5 days post symptom onset. The simplest relationship considered was linear, while the most complex was non-linear. No *a priori* assumption was made about the shape of the relationship. Other researchers found a general decline in the probability of transmission with day of illness [16], but because the current study participants were enrolled earlier in their viremic period, it was unclear if the patterns previously observed would translate directly to this setting. In terms of the non-linear functional form, two approaches were considered: (1) a smooth non-linear b-spline-based term and (2) a fully non-parametric functional form where each day of illness had its own effect fit. Ostensibly, the smooth term would be the more parsimonious approach while the non-parametric functional form could allow for direct comparison between different days' effects.

**Extrinsic incubation.** The duration of extrinsic incubation ranged from 7 to 16 days. *A priori* it was assumed that the probability of transmission would increase for longer incubation times. As such, the most complex functional form considered was a shape-constrained non-decreasing, monotonic smooth relationship. By design, experiments were either stopped early (7–8 days) or allowed to continue beyond EIP when most transmission is expected to occur

naturally (12–16 days). There was no sampling on days 9, 10, or 11 post blood feeding. In addition to the complex model described above, there were two simpler models considered, as well: (1) a linear relationship for incubation time or (2) a stepped relationship with one constant effect for 7–8 days of incubation and a different constant effect for 12–16 days of incubation. Both of these simpler models use a single degree of freedom, so in the absence of fit dominance, it would be difficult to identify the most parsimonious between them.

**Repeated measures.** Of the 27 participants included in the analysis, 9 contributed to only one experiment; i.e., a single feeding event resulting in a single incubation-day of illness combination. Fifteen contributed to two experiments, one contributed to three, and two contributed to four. While there is potential for the influence of repeated measures on the experiment outcome (e.g., a participant who is systematically more likely to infect mosquitoes than their viremia would indicate), there are relatively low numbers of total participants and repeatedly measured participants. Due to low sample sizes and concerns around convergence and interpretability, a random effect by participant was not included. Care must be taken to not overinterpret the conclusions of our analysis as the sample size is low.

## Results

We identified people with active infections by detecting viral RNA in their blood, which we refer to as viremia hereafter. A total of 54 viremic people consented to blood feed mosquitoes directly 1–4 times during their infection, which resulted in 116 mosquito feeding events. All 54 study participants were symptomatic; 96.3% were febrile. Forty-three (79.6%) participants were infected with DENV-2, 2 (3.7%) with DENV-3, and 9 (16.7%) with ZIKV. Our analysis excluded events for which viremia was detectable at enrollment but undetectable at the time of mosquito feeding, which was the case for all events involving participants with DENV-3 and ZIKV infections. Five participants (2 DENV-3 and 3 DENV-2 infections) were infectious to mosquitoes despite having undetectable viral RNA in their blood at the time of feeding. In total, our analysis was conducted on 50 feeding events involving 27 participants (44.4% female) that had detectable DENV-2 viremia at the time of feeding. Of these feeding events, 58% were on participants ≤17 years of age, 30% on participants 18–31 years of age, and 12% on participants 40–80 years of age (S1 Fig). Corresponding primary data are provided in supplementary information (S1 File) and their descriptive statistics are shown in Table 1. Variation in the number of mosquitoes tested reflects variation in the number of mosquitoes available, variation in the blood feeding rate and/or variation in the mosquito survival rate during the incubation period.

We analyzed the probability of a mosquito becoming infectious (i.e., a direct measure of vector competence) using logistic regression models. The final selected model retained a linear effect of viremia, a non-linear smooth effect of day of illness, and a stepped effect of extrinsic

**Table 1. Summary table of descriptive statistics.** Summary statistics are provided for the main variables of the final dataset analyzed.

| Variable | Range | Median | Mean | Std. dev. |
|---|---|---|---|---|
| Participant age (years) | 5–80 | 17 | 22.4 | 17.0 |
| Days of illness at the time of feeding | 1–5 | 2 | 2.46 | 1.20 |
| Number of feeding events per participant | 1–4 | 2 | 1.85 | 0.82 |
| Viremia at the time of feeding ($\log_{10}$ scale) | 2.0–9.1 | 5.63 | 5.29 | 2.05 |
| Number of mosquitoes tested | 1–20 | 18.5 | 16.6 | 4.84 |
| Percentage of infectious mosquitoes | 0–45 | 0 | 8.76 | 13.6 |

**Table 2. Parameters of the final selected model.** The probability of a mosquito being infectious was analyzed using logistic regression models. The table shows the effects retained in the final selected model. Eff. df: effective degrees of freedom; Ref. df: reference degrees of freedom.

| Model term | Estimate | Std. error | z-value | p-value |
|---|---|---|---|---|
| Intercept | -7.0656 | 0.7934 | -8.905 | $<2\times10^{-16}$ |
| Log viremia | 0.3237 | 0.0831 | 3.895 | 0.000098 |
| Incubation > 11 days | 3.0541 | 0.6063 | 5.038 | 0.00000047 |
| **Model term** | **Eff. df** | **Ref. df** | **Chi²** | **p-value** |
| Day of illness | 1.91 | 1.99 | 9.775 | 0.00565 |

incubation time. While this model was not the best model in terms of AIC, no model had an AIC less than 2 units smaller and all models with smaller AIC were deemed biologically implausible (see S1 Text for details). Each term was statistically significant (Table 2). For each $\log_{10}$-unit increase in viremia, the odds of a mosquito being infectious were 1.38 times higher (odds ratio [OR] = 1.38, *p*-value <0.0001, 95% confidence interval [CI]: 1.17–1.63). Extrinsic incubation for 12 days or greater increased the odds of a mosquito being infectious by 21.3 times relative to 7 or 8 days of incubation (OR = 21.3, *p*-value <0.00001, 95% CI: 6.46–69.58). Fig 1 shows ORs for each day of illness against the mean estimated odds for day of illness of 1.

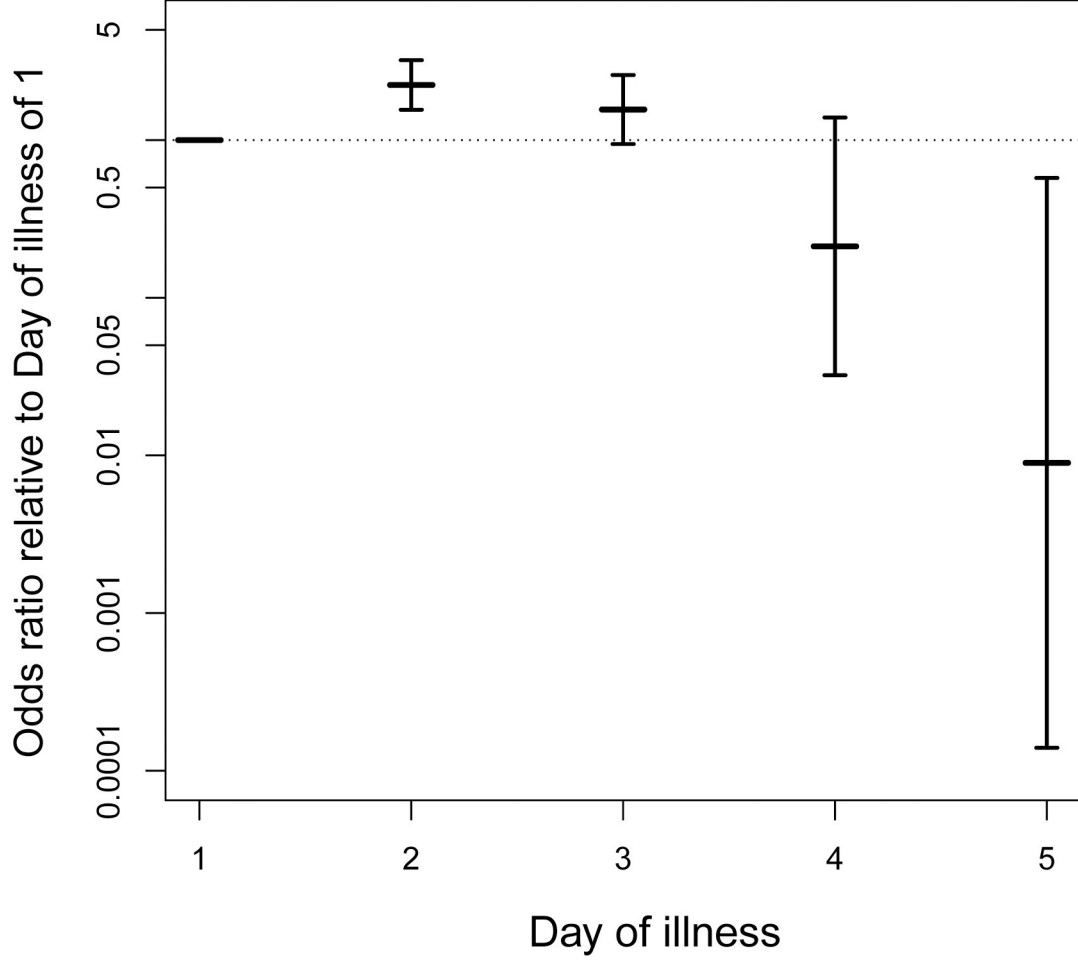

**Fig 1. DENV transmission probability is highest on the second day of illness.** The odds of a mosquito being infectious for each day of illness are plotted on a $\log_{10}$ scale relative to the mean estimated odds for the first day of illness. Vertical bars show the 95% confidence intervals of the odds ratios. The horizontal dotted line represents an odds ratio of 1.

Of particular note, the ORs for the second day of illness compared to the mean estimated effect for the first day of illness was 2.24 (95% CI: 1.56–3.21). This approach does not easily produce a *p*-value for this comparison, but the entire non-linear smooth model is statistically significant (*p*-value = 0.00565). Another model that resulted in a similar fit does allow for a direct calculation of the OR between the first and second days of illness. The model with no smooth terms, a linear effect of viremia, a non-parametric effect of day of illness, and a stepped effect of incubation time had an AIC of 139.52 ($\Delta AIC$ = 3.34). That model, which is arguably simpler even though it uses more degrees of freedom, revealed extremely similar effects of viremia and incubation time. In that model, which fit the effect of each day of illness as its own coefficient, the OR for the second day of illness compared to the first day of illness was 2.23 (*p*-value = 0.0062, 95% CI: 1.26–3.96). Fig 2 illustrates the fitted effects of the covariates on predicted probabilities across a wide range of potential values.

## Discussion

In this study, we quantified human-to-mosquito DENV transmission from viremic people to determine the drivers of infectiousness to mosquitoes in a natural context. Our experimental setup allowed us to blood feed mosquitoes directly on human volunteers with mild illness during their DENV infection [9,15,32]. We measured the proportion of mosquitoes that became infectious (i.e., that had infectious virus in their saliva) and used regression analyses to identify the factors influencing this process. We confirmed that the magnitude of viremia is a significant predictor of the probability of a blood-fed mosquito becoming infectious. Consistent with earlier studies, a longer extrinsic incubation also increased transmission probability [46]. A result that is different from previous studies was that the day of illness influenced the probability of onward DENV transmission in a non-monotonic fashion with a maximum around 2 days after symptom onset. This finding challenges the previous belief that the maximum level of infectiousness to mosquitoes coincides with symptom onset [17].

Day of illness was previously found to negatively influence human-to-mosquito transmission of DENV-1 and DENV-2 from hospitalized patients enrolled 2–4 days after symptom onset [16]. For each unit increase in day of illness, the odds of a mosquito becoming infected were 0.6 times lower [16]. Our results reveal that the relationship between infectiousness and day of illness is more complex than a simple linear relationship. By using non-linear functional forms in our regression models, we found that the odds of DENV transmission on the second day of illness were 2.23 times higher than on the first day of illness. Infectiousness subsequently decreased until the fifth day of illness. One important implication of this finding is that virus transmission can be substantially reduced if mildly symptomatic DENV-infected people take protective measures against mosquito bites within the first two days of their illness. Protective measures could include personal protection (e.g., topical repellents) and/or insecticide spraying at the locations of daytime activity.

At this point, we can only speculate on the factors other than viremia that drive temporal variation in DENV infectiousness to mosquitoes. DENV-infected people experience most symptoms during their first week of illness, but there is substantial heterogeneity in the duration and reported intensity of individual symptoms [47]. Immune factors, metabolites or changes in blood concentration of viral antigens during infection could modulate mosquito infection [16,26,27,29,30]. The effect of human blood factors on vector competence could be due to their direct influence on mosquito-virus interactions or indirectly mediated by changes of the mosquito gut microbiota [48].

A limitation of our study is the small number of human participants that contributed to the analysis and the diminishing number of volunteers who participated in repeated feedings over

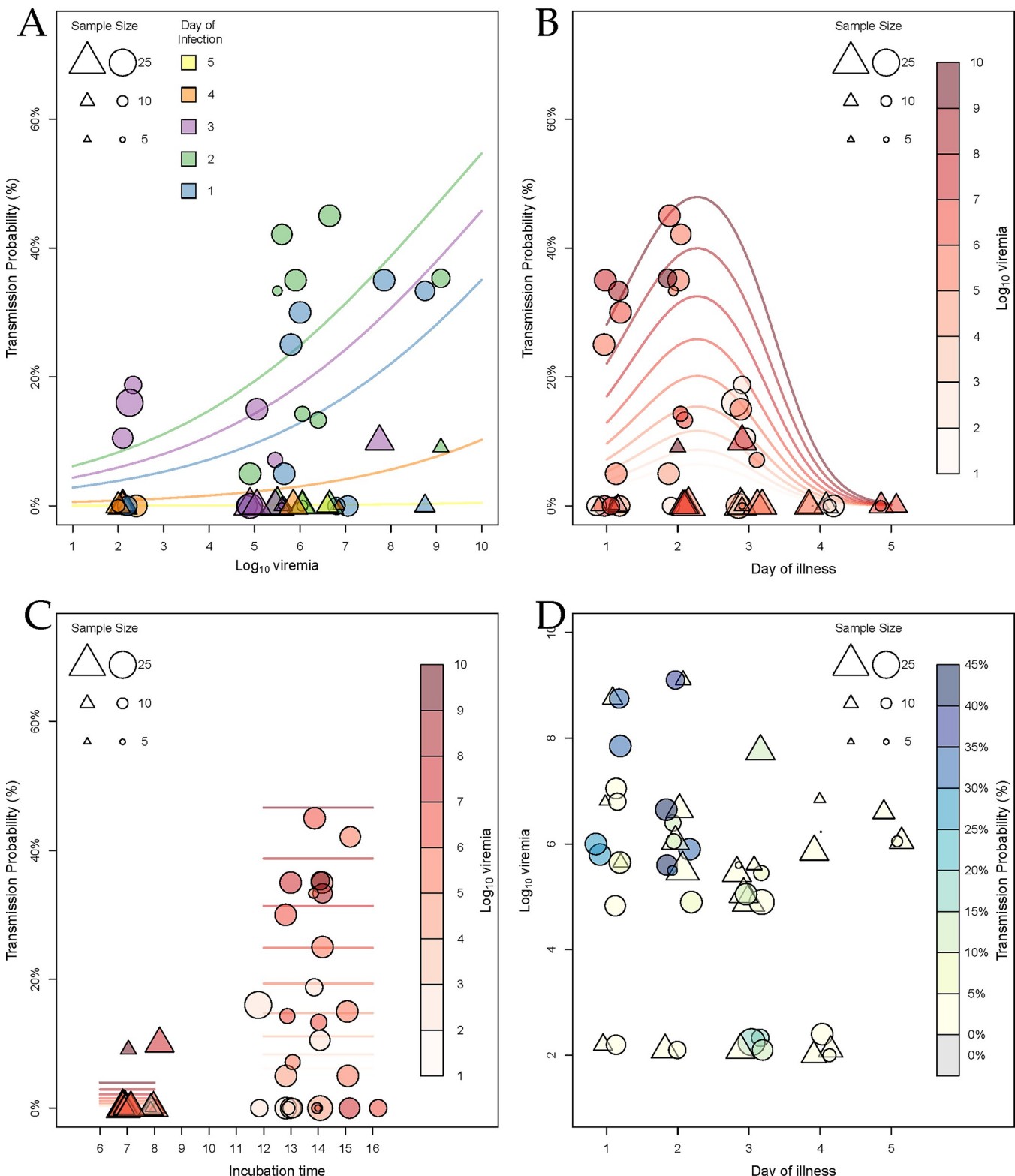

**Fig 2. Human-to-mosquito DENV transmission depends on viremia, incubation time and day of illness.** DENV transmission probability was estimated by the proportion of blood-fed mosquitoes that had infectious virus in their saliva. Each circle (12–16 days of incubation) or triangle (7–8 days of incubation) represents a single mosquito feeding event and the size of the circle or triangle is proportionate to the number of mosquitoes analyzed. Lines represent the regression fits to the data. Lines and data points are color and shape coded according to the inset scale. (a) Transmission probability as a function of $log_{10}$-

transformed viremia. (b) Transmission probability as a function of day of illness. (c) Transmission probability as a function of extrinsic incubation time in days. (d) $Log_{10}$-transformed viremia as a function of day of illness.

time. We excluded 2 participants infected with DENV-3, all 9 participants infected with ZIKV, and 16 of the 43 participants with a DENV-2 infection from the final dataset because they did not have detectable viremia at the time of mosquito feeding. Due to the low number of participants retained in the analysis (n = 27), we could not account for a random effect of participant and our statistical power was limited. In addition, participants in our study did not represent the entire spectrum of disease manifestations. All of our study participants were mildly symptomatic, which is the case for the majority of DENV infections [5]. It would be helpful in the future to determine whether our conclusions extend to the entire disease spectrum, ranging from asymptomatic infections to people that experience severe dengue, and to DENV serotypes other than DENV-2.

An additional limitation of our study was to measure viremia as the concentration of viral RNA in blood, which is distinct from the concentration of infectious viral particles [49]. Variation in infectiousness to mosquitoes could reflect differences in the ratio of infectious particles over viral genome copies. Intriguingly, a few participants with undetectable viremia were infectious to mosquitoes, as was observed in our previous study [7]. This could reflect variation in viral titer between venous blood (used to measure viremia) and capillary blood (imbibed by mosquitoes) and/or between serum and whole blood.

It is notable that despite conducting feeding experiments on 9 participants infected with ZIKV, none were infectious to mosquitoes. Viremia was undetectable on the day of feeding for all ZIKV-infected participants, including four feeding experiments that took place as early as two days post symptom onset. This is consistent with the previous observation that ZIKV viremia post symptom onset can be too low to achieve a productive mosquito infection [50]. It is possible that effective human-to-mosquito ZIKV transmission primarily occurs prior to symptom onset. Additional studies are required to determine the kinetics of human-to-mosquito ZIKV transmission during the viremic period.

Our results shed new light on variation within and between different people in their contribution to onward DENV transmission. Our finding that mildly symptomatic people are most infectious to mosquitoes on the second day of their illness points to an overlooked aspect of the dynamics of DENV infectiousness to mosquitoes during the viremic period. The net contribution of an individual to DENV transmission is a function of infectiousness dynamics and the frequency of human-mosquito contacts. Exposure to mosquitoes is heterogeneous [51–53] and difficult to quantify due to the unknown effect of coupled heterogeneities [54]. Human-mosquito contacts also depend on symptom severity [55], which highlights the need to relate illness with infectiousness dynamics. Detailed longitudinal information at the individual level can be used to improve surveillance and outbreak response as well as help to better inform modeling assessments of DENV transmission dynamics and the relative public health impact of different intervention tools and strategies [9].

## Supporting information

**S1 Fig. Demographic characteristics of 50 DENV-2 viremic feedings evaluated for infectiousness to _Aedes aegypti_ mosquitoes.**
(TIF)

**S1 File. Primary data set used for analysis.**
(CSV)

**S1 Text. Model Selection Procedures.** Contains Figs A-B and Tables A-C. **Fig A.** Flow chart of model selection process. **Fig B.** Linear model fitting for day of illness. The plot shows the residuals of the linear fit against day of illness. Thin lines represent the 25[th] and 75[th] percentiles and solid lines represent the median. **Table A.** GLM with linear terms for each covariate (AIC = 163.2932). **Table B.** GAM with nonlinear smooth term for day of illness and linear terms otherwise (AIC = 146.0056). Eff. df: effective degrees of freedom; Ref. df: reference degrees of freedom. **Table C.** GLM with day of illness as a factor and linear terms otherwise (AIC = 149.6279).
(DOCX)

## Acknowledgments

We thank the residents of Iquitos for their support and participation in this study. We are grateful to the Loreto Regional Health Department, including Drs. Hugo Rodriguez-Ferruci, Christian Carey, Carlos Alvarez, Hernan Silva and the Lic. Wilma Casanova Rojas, who all facilitated our work in Iquitos. W. Lorena Quiroz and Jhonny Cordova conducted dengue positive case follow-up and interviews and acknowledge the NAMRU-6 Iquitos febrile surveillance and laboratory teams in the rapid capture and diagnosis of dengue and Zika cases for direct mosquito feeding studies. We acknowledge the NAMRU-6 Virology and Emerging Infectious Disease and Command during the execution of this project. A special thanks to Gloria Talledo for her ongoing support with the preparation of protocols and reports for this project. We appreciate the commentary and advice provided by the NAMRU-6 Institutional Review Board and Research Administration Program for the duration of the study.

## Author Contributions

**Conceptualization:** Louis Lambrechts, Robert C. Reiner, Jr, Kanya C. Long, T. Alex Perkins, Alun L. Lloyd, Lance A. Waller, Uriel Kitron, Sarah A. Jenkins, Robert D. Hontz, Lauren B. Carrington, Cameron P. Simmons, John P. Elder, Valerie A. Paz-Soldan, Gonzalo M. Vazquez-Prokopec, Alan L. Rothman, Christopher M. Barker, Thomas W. Scott, Amy C. Morrison.

**Data curation:** Robert C. Reiner, Jr, M. Veronica Briesemeister, Patricia Barrera, William H. Elson, Alfonso Vizcarra, Anna B. Kawiecki, Alan L. Rothman, Christopher M. Barker, Amy C. Morrison.

**Formal analysis:** Louis Lambrechts, Robert C. Reiner, Jr, William H. Elson, T. Alex Perkins, Alun L. Lloyd, Gonzalo M. Vazquez-Prokopec, Alan L. Rothman.

**Funding acquisition:** Louis Lambrechts, Robert C. Reiner, Jr, T. Alex Perkins, Alun L. Lloyd, Uriel Kitron, John P. Elder, Valerie A. Paz-Soldan, Gonzalo M. Vazquez-Prokopec, Alan L. Rothman, Thomas W. Scott, Amy C. Morrison.

**Investigation:** Louis Lambrechts, M. Veronica Briesemeister, Patricia Barrera, William H. Elson, Alfonso Vizcarra, Helvio Astete, Isabel Bazan, Crystyan Siles, Stalin Vilcarromero, Mariana Leguia, Anna B. Kawiecki, Sarah A. Jenkins, Robert D. Hontz, J. Sonia Ampuero, Valerie A. Paz-Soldan, Gonzalo M. Vazquez-Prokopec, Alan L. Rothman, Thomas W. Scott, Amy C. Morrison.

**Methodology:** Louis Lambrechts, Robert C. Reiner, Jr, M. Veronica Briesemeister, Patricia Barrera, Kanya C. Long, Mariana Leguia, T. Alex Perkins, Alun L. Lloyd, Uriel Kitron, Sarah A. Jenkins, Robert D. Hontz, Wesley R. Campbell, Lauren B. Carrington, Cameron P.

Simmons, John P. Elder, Gonzalo M. Vazquez-Prokopec, Alan L. Rothman, Christopher M. Barker, Thomas W. Scott, Amy C. Morrison.

**Project administration:** Sarah A. Jenkins, Robert D. Hontz, Wesley R. Campbell, Gisella Vasquez, Valerie A. Paz-Soldan, Thomas W. Scott, Amy C. Morrison.

**Resources:** Louis Lambrechts, Sarah A. Jenkins, Robert D. Hontz, Wesley R. Campbell, Thomas W. Scott, Amy C. Morrison.

**Software:** Christopher M. Barker.

**Supervision:** M. Veronica Briesemeister, Patricia Barrera, Helvio Astete, Isabel Bazan, Crystyan Siles, Stalin Vilcarromero, Mariana Leguia, Sarah A. Jenkins, Robert D. Hontz, J. Sonia Ampuero, Valerie A. Paz-Soldan, Thomas W. Scott, Amy C. Morrison.

**Visualization:** Robert C. Reiner, Jr.

**Writing – original draft:** Louis Lambrechts, Robert C. Reiner, Jr, Kanya C. Long, Valerie A. Paz-Soldan, Thomas W. Scott, Amy C. Morrison.

**Writing – review & editing:** Louis Lambrechts, Robert C. Reiner, Jr, M. Veronica Briesemeister, Patricia Barrera, Kanya C. Long, William H. Elson, Alfonso Vizcarra, Helvio Astete, Isabel Bazan, Crystyan Siles, Stalin Vilcarromero, Mariana Leguia, Anna B. Kawiecki, T. Alex Perkins, Alun L. Lloyd, Lance A. Waller, Uriel Kitron, Sarah A. Jenkins, Robert D. Hontz, Wesley R. Campbell, Lauren B. Carrington, Cameron P. Simmons, J. Sonia Ampuero, Gisella Vasquez, John P. Elder, Valerie A. Paz-Soldan, Gonzalo M. Vazquez-Prokopec, Alan L. Rothman, Christopher M. Barker, Thomas W. Scott, Amy C. Morrison.

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
