## [Decision Letter · Decision Letter 0]

12 May 2023

Dear Dr. Morrison,

Thank you very much for submitting your manuscript "Direct mosquito feedings on dengue and Zika virus-infected people reveal dynamics of human infectiousness" for consideration at PLOS Neglected Tropical Diseases. As with all papers reviewed by the journal, your manuscript was reviewed by members of the editorial board and by several independent reviewers. In light of the reviews (below this email), we would like to invite the resubmission of a significantly-revised version that takes into account the reviewers' comments. 

The reviewers and editor appreciate that the central finding of the paper, where human infectiousness to mosquitoes via bloodmeals continues to increase after (mild) dengue symptoms begin, is original and advances knowledge in the field. The authors highlight that mild dengue infections tend to be challenging to detect by surveillance and as a result their role in community transmission has been understudied and underappreciated. While the analysis and conclusions are based on a relatively limited sample size of participants eligible for and completing the bloodmeal feedings, the authors have considered appropriate statistical procedures of model selection for parameter inference, though they might consider adding a summary table of descriptive statistics, implementing alternative error distributions for modeling the data, and clarifying further the steps of their model selection using a supplemental table or image. Several suggestions were made and should be considered to improve the clarity of results shown in Figure 2. The authors adequately advise caution with the data, results, and inference, including statistical power limitations. However, reviewer concerns which must be addressed for clarity include the techniques used for mosquito husbandry and rearing, intrathoracic inoculation, and blood feeding on humans, and the potential influence of the techniques used on the results and conclusions. Lastly, the authors may consider revising the article title to more accurately reflect the focus of the results and discussion, i.e. specifically on dengue-2.

We cannot make any decision about publication until we have seen the revised manuscript and your response to the reviewers' comments. Your revised manuscript is also likely to be sent to reviewers for further evaluation.

Sincerely,

Amy T. Gilbert

Academic Editor

Elvina Viennet

Section Editor

The reviewers and editor appreciate that the central finding of the paper, where human infectiousness to mosquitoes via bloodmeals continues to increase after (mild) dengue symptoms begin, is original and advances knowledge in the field. The authors highlight that mild dengue infections tend to be challenging to detect by surveillance and as a result their role in community transmission has been understudied and underappreciated. While the analysis and conclusions are based on a relatively limited sample size of participants eligible for and completing the bloodmeal feedings, the authors have considered appropriate statistical procedures of model selection for parameter inference, though they might consider adding a summary table of descriptive statistics, implementing alternative error distributions for modeling the data, and clarifying further the steps of their model selection using a supplemental table or image. Several suggestions were made and should be considered to improve the clarity of results shown in Figure 2. The authors adequately advise caution with the data, results, and inference, including statistical power limitations. However, reviewer concerns which must be addressed for clarity include the techniques used for mosquito husbandry and rearing, intrathoracic inoculation, and blood feeding on humans, and the potential influence of the techniques used on the results and conclusions. Lastly, the authors may consider revising the article title to more accurately reflect the focus of the results and discussion, i.e. specifically on dengue-2.

Reviewer's Responses to Questions

**Key Review Criteria Required for Acceptance?**

**Methods**

-Are the objectives of the study clearly articulated with a clear testable hypothesis stated?

-Is the study design appropriate to address the stated objectives?

-Is the population clearly described and appropriate for the hypothesis being tested?

-Is the sample size sufficient to ensure adequate power to address the hypothesis being tested?

-Were correct statistical analysis used to support conclusions?

-Are there concerns about ethical or regulatory requirements being met?

Reviewer #1: The methods are sane, I just recommend making the model selection procedure a bit clearer through a graph or a table (as a supplement).

Reviewer #2: -Are the objectives of the study clearly articulated with a clear testable hypothesis stated?

yes

-Is the study design appropriate to address the stated objectives?

yes

-Is the population clearly described and appropriate for the hypothesis being tested?

yes

-Is the sample size sufficient to ensure adequate power to address the hypothesis being tested?

yes

-Were correct statistical analysis used to support conclusions?

I suspect - these are outside of my expertise.

-Are there concerns about ethical or regulatory requirements being met?

no

Reviewer #3: Major issues: 

Variability across data collection: there is substantial variation in the number of mosquitoes assessed in this data set, yet no explanation for this is provided. Mosquitoes rarely die off in large number during such experiments unless they are improperly treated. This may be the case here as the methods describe knock down of 5 min at -20C, which is long even for temperate mosquitoes. 30 sec at -20C is typically sufficient for Aedes aegypti strains and much longer tends to result in death. Therefore, often they are knocked down at 4C instead or -20C. Was this incorrectly provided or is this GDLS particularly cold hearty? Alternatively, were the mosquitoes not properly prepared to blood feed? For the mosquitoes that were collected post 12 days the range of collection varies from 12-16 days this is a large range that adds further complexity in analysis on an already small sample set. This adds further to the lack of confidence in the data set. Then to assess infectiousness, saliva samples are collected directly in a pipet tip and intrathoracically injected into 5 new mosquitoes seeming directly from the pipet tip. Intrathoracic injections with a pulled needle in Aedes species are difficult enough to do without killing any mosquitoes, this reviewer would like to know what ratio remain alive given it would potentially substantially skew the transmission probability here. Is day two only the highest probability because there were the most individuals available for assessment on this day and, with the highest viremia individual present on this day and counted twice as a result of the combination of the pre 12-day incubation set? Why do the authors not show the RT-qPCR results, the original viremia results, infect cell culture cells rather than this complex mosquito infection process with all the error it introduces. Or if cell culture isn’t possible then run RT-qPCR on the saliva directly from the original set of mosquitoes. 

Statistical analyses: there are no statistics to demonstrate whether any of the data are reliable as the only statistics shown were identified for the purpose of being most statistically significant. There are no statistics shown for Figure 1 and what the error bars represent is not included, and day 1-3 seem nearly identical given the small numbers in the study.

**Results**

-Does the analysis presented match the analysis plan?

-Are the results clearly and completely presented?

-Are the figures (Tables, Images) of sufficient quality for clarity?

Reviewer #1: The results are clear but Figure 2 could be improved in my opinion. I make some suggestions in my general comments.

Reviewer #2: -Does the analysis presented match the analysis plan?

yes

-Are the results clearly and completely presented?

yes

-Are the figures (Tables, Images) of sufficient quality for clarity?

yes

Reviewer #3: Major issues:

In the main figure of the paper, Figure 2, the legend states that a and b are only the mosquito feeding events that were incubated for between 12-16 days, which would be appropriate. However, this is not accurate. The easiest to identify is the highest viremia individuals from day two. Both extrinsic incubations are present in a, b, and d. Given 7-8 days are largely irrelevant EI time points this is inappropriate and even if it were appropriate, it doesn’t match the description provided by the authors. Overall, the inattention to detail is a cause for concern across the data provided. In addition to correcting this inaccuracy the data overall should be presented in a way that the reader can fully access. By combing four assessments per graph with little explanation for the relevance of each aspect it is difficult to assess what is significant and what is noise. Unless it’s mostly noise.

The authors bled and fed mosquitoes on at minimum 50% children. Children are not miniature adults and respond differently to viral infections when compared with adults. Given the small sample size in this manuscript it serves primarily as a steppingstone/pilot to design further directed studies based on the observations provided. The authors should provide as much detail about the data acquired as possible. Instead, the data are highly condensed and filtered.

**Conclusions**

-Are the conclusions supported by the data presented?

-Are the limitations of analysis clearly described?

-Do the authors discuss how these data can be helpful to advance our understanding of the topic under study?

-Is public health relevance addressed?

Reviewer #1: The conclusion and discussion are well articulated and do a good job of positioning the study and pointing to possible next steps.

Reviewer #2: -Are the conclusions supported by the data presented?

yes

-Are the limitations of analysis clearly described?

yes (sample size)

-Do the authors discuss how these data can be helpful to advance our understanding of the topic under study?

yes, very clearly, and with strong implications for disease control

-Is public health relevance addressed?

yes

Reviewer #3: (No Response)

**Editorial and Data Presentation Modifications?**

Reviewer #1: (No Response)

Reviewer #2: This is a great piece of work and I only have a few comments.

page 5 - starting here there are a few references to this being a field-based study, but it seems like a lot of this was done in the laboratory or insectary. It struck me as a little odd, though I don't disagree altogether.

page 8 - "As the mosquitoes emerged, females were separated into 1-pint cardboard cages containing 25 Ae. aegypti each." Were they not allowed to mate?

page 8 - in your methods under "Blood feeding" - I think the incubation periods allowed for mosquitoes after the feed could be clarified. I only understood this after getting through more of the manuscript. When you say, for example, "Experiments were either stopped early (7-8 days) or late (12-16 days)...", what are you considering an experiment? I would try something more explicit here like "~X mosquitoes were harvested on days X, X, X, and X for analysis of vector competence".

Reviewer #3: Title: no dynamics are revealed for ZIKV, or DENV3. The title is misleading. Either add a substantial amount of information about the ZIKV and DENV3 subjects/samples or focus the paper on DENV-2.

**Summary and General Comments**

Reviewer #1: Note : please provide line numbers in the revised version of the manuscript

Introduction

The introduction is well written. As I understand it, the dataset you’re using comes from studies already published (refs 9, 15, 32). It would be helpful to more clearly identify what were the key messages of these previous studies and how the current one comes to complete the picture.

Also, can you put more emphasis on the differences and similarities between your study and the ones done in Vietnam and Cambodia (Nguyen et al. 2013, Duong et al. 2015)? Did you have a priori hypotheses on the factors that could drive differences between your results and theirs (study location, viral strains, people past exposure etc.) or were you expecting this to mostly replicate their observations and increase the robustness of the conclusions? 

Lastly, an important added value of your study in my opinion is to have repeated mosquito feedings on the same individuals, which was not the case in Nguyen et al. 2013 and Duong et al. 2015 if I’m not mistaken. This is implied when you mention the “temporal dynamics of infectiousness”, or that you want to understand “variation within people”, but it could be stated even more clearly.

Methods

* Vector competence

You mention that “If the pool of RNA extracts was negative, the participant was considered to be not infectious to mosquitoes and no further mosquito testing was performed.” I guess you mean no further testing for this particular feeding on a given individual, and not possible subsequent feedings? Please precise.

* General model considerations

You mention using logistic regressions with a binomial link function. First, I think it would be more accurate to say a binomial error distribution with a logit (I assume as it’s the most common) link function. And second, have you tried comparing with a betabinomial error distribution? This would account for overdispersion, and in my experience often gives the best fit for this kind of data (improvement more tangible than including random effects).

Please precise that analyses were conducted in R before you start mentioning package names, and not after.

I acknowledge that authors intend to share their code and data through open repositories once the paper is accepted, which is great (and I intend to look at your code once it's there, as this is of interest to me).

* Viremia

You mention RNA concentration (and state in the discussion that this is a limit because it is distinct from infectious viral particles) but the initial paper (ref 9) also mentions fluorescent focus assay in its Figure 3. This would importantly give information on infectious viral load, particularly an estimation of the ratio of infectious over total viral particles, which is rare in the literature. Why was this not used here?

* Repeated measures

You explain not including a random effect by participant. But the way you phrase it, I don’t understand if you tried and it did not work/was not prefered over the model not including random effects, which I would understand, or if you simply chose not to try, which I would find odd (although as said before, I suspect trying a betabinomial model is more likely to improve your fit than random effects for this kind of experimental design).

Overall, I find the approach regarding data analysis very sane, but I’m still unsure that I understand perfectly what has been done, in terms of all model forms that were compared. As the supporting information currently only gives additional details on the results of the selection procedure, I still think it would be helpful to detail the framework of your model comparison procedure with a graph or a table (as a supplementary item as well). This would be a didactic opportunity for young (and sometimes not so young) researchers facing the same kind of analysis, and (hopefully) ensure that stepwise regression is no longer the “go-to” analysis.

Results

I am of course intrigued by your participants with no detectable viremia that did transmit to mosquitoes. Do you think this might be something to investigate, or do you attribute this to anecdotal evidence? And why did you exclude those datapoints (for DENV-2) from model fitting : did you want to constrain your probability to zero for absence of viremia? If yes, could you not have assign these participants a viremia value at the limit of detection? By the way it does not seem that the limit of detection of your assay is reported anywhere so I would add this information.

Figure 2 : y-axis text (not title) should be horizontal for better readability for all sub-figures. Same goes for the “Day of infection” legend title in a), which could then be put on top of the discrete color scale. Same with Log viremia for b-d (and it should log10 rather than log).

In the caption of the figure : “with after”, when describing (a), should be “after”

I find the figure a bit confusing to read, here are a few suggestions/comments to improve it :

For a and b : add transparency to the points and maybe change the color scales because it’s hard to associate the points with the curves of the same values, particularly for mid-range values. Maybe two types of viridis scales for a and b (https://ggplot2.tidyverse.org/reference/scale_viridis.html) ? At least for the days of infection as you only have 5 levels it should help distinguish values.

For c : why not drop the sample size and use point size to show day of illness ? Maybe it’ll get confusing with your fitted lines… But at least I don’t see why you would show the fits per viremia and not per day of illness as well : maybe a supplementary figure?

For d : the color and the y axis are redundant, why not use the color for transmission probability? Not sure sample size is really needed as well, but points shape could show the two categories of incubation time? Just a thought...

Discussion

The discussion is well structured and addresses the main limitations, contributions, and perspectives clearly.

Reviewer #2: I have just one final general comment. There will certainly be some researchers reading this that don't appreciate seeing more advanced statistical techniques. They might even gripe. I would appease this by adding in a table with some sort of summary of the raw data. The odds-ratio plot comes closest to this. They may also not be github users, so although the raw data will be available there, a summary would go a long way.

Reviewer #3: In the manuscript entitled “Direct mosquito feedings on dengue and Zika virus-infected people reveal dynamics of human infectiousness” by Louis Lambrechts and Robert C Reiner Jr. et al, the authors present data linking max transmission probability to the day of illness for dengue-2 infections. Specifically, the data suggest day two post symptom onset for dengue-2 may result in the highest transmission probability increasing from day one and falling through day five. This work involved a great deal of logistics to enroll participants and collect the samples, however the methods were previously published, so that aspect is not a new contribution to the field. All the data presented in the manuscript fit into two figures and a table, with Figure 1 and Figure 2b being the same data analyzed in different ways. While the finding that human infectiousness to mosquitoes during bloodmeals continues to increase after symptoms begin is novel, there are many issues that need to be addressed and this work seems less than what should be a minimal publishable unit.

PLOS authors have the option to publish the peer review history of their article (what does this mean?). If published, this will include your full peer review and any attached files.

Reviewer #1: No

Reviewer #2: No

Reviewer #3: No
---

## [Decision Letter · Decision Letter 1]

14 Aug 2023

Dear Dr. Morrison,

We are pleased to inform you that your manuscript 'Direct mosquito feedings on dengue-2 virus-infected people reveal dynamics of human infectiousness' has been provisionally accepted for publication in PLOS Neglected Tropical Diseases.

Best regards,

Elvina Viennet, PhD

Section Editor

Thank you for your thorough responses to reviewers. Please find attached few minor comments.

Reviewer's Responses to Questions

**Key Review Criteria Required for Acceptance?**

**Methods**

-Are the objectives of the study clearly articulated with a clear testable hypothesis stated?

-Is the study design appropriate to address the stated objectives?

-Is the population clearly described and appropriate for the hypothesis being tested?

-Is the sample size sufficient to ensure adequate power to address the hypothesis being tested?

-Were correct statistical analysis used to support conclusions?

-Are there concerns about ethical or regulatory requirements being met?

Reviewer #1: Yes to all.

Small typo, l.348 (version without tracked changes) : logit not logic

In file S.1, l.73 : "As both a linear and a non-linear…" sounds odd, I think “With either a linear or a non-linear…” would be better?

**Results**

-Does the analysis presented match the analysis plan?

-Are the results clearly and completely presented?

-Are the figures (Tables, Images) of sufficient quality for clarity?

Reviewer #1: Yes to all.

In Figure 2C, I wonder why fitted lines per viremia levels are the same for both EIP categories. Shouldn't the longer EIP had higher transmission probability, for a given viremia?

In Figure S2 : what does the thick and thin lines represent for each day?

Table S2 : consider providing more detailed column information (e.g by turning this file into an Excel with a separate sheet for column information), even though the headers are quite explicit, to avoid any confusion

**Conclusions**

-Are the conclusions supported by the data presented?

-Are the limitations of analysis clearly described?

-Do the authors discuss how these data can be helpful to advance our understanding of the topic under study?

-Is public health relevance addressed?

Reviewer #1: Yes to all

**Editorial and Data Presentation Modifications?**

Reviewer #1: In the authors' summary :

l.128 DENV-3 instead of DENV-2

l.129 one to four times? You say later that some participants were fed on four times

l.138-139 “people with mild illness are thought to contribute more to DENV transmission than more severely ill people” : this is counterintuitive at first so as you cannot cite the study by Bosch et al. Here, maybe give a bit more information to understand this statement

l.142 a better understanding

**Summary and General Comments**

Reviewer #1: (No Response)

PLOS authors have the option to publish the peer review history of their article (what does this mean?). If published, this will include your full peer review and any attached files.

Reviewer #1: No

---

## [Editor Report · Acceptance letter]

24 Aug 2023

Dear Dr. Morrison,

We are delighted to inform you that your manuscript, "Direct mosquito feedings on dengue-2 virus-infected people reveal dynamics of human infectiousness," has been formally accepted for publication in PLOS Neglected Tropical Diseases.

Best regards,

Shaden Kamhawi

co-Editor-in-Chief

Paul Brindley

co-Editor-in-Chief
